# Isolation and Characterization of Antioxidant Peptides from Dairy Cow (*Bos taurus*) Placenta and Their Antioxidant Activities

**DOI:** 10.3390/antiox13080913

**Published:** 2024-07-29

**Authors:** Xinyu Tian, Zeru Zhang, Yuquan Zhao, Anguo Tang, Zhi Zeng, Weijian Zheng, Hanwen Zhang, Yuxin Luo, Wei Lu, Lei Fan, Liuhong Shen

**Affiliations:** 1The Key Laboratory of Animal Disease and Human Health of Sichuan Province, The Medical Research Center for Cow Disease, College of Veterinary Medicine, Sichuan Agricultural University, Chengdu 611130, China; 2021303134@stu.sicau.edu.cn (X.T.); zhangzr@stu.sicau.edu.cn (Z.Z.); zhaoyuquan@stu.sicau.edu.cn (Y.Z.); tangag@haid.com.cn (A.T.); 2023303092@stu.sicau.edu.cn (Z.Z.); zhengweijian@stu.sicau.edu.cn (W.Z.); 2023203075@stu.sicau.edu.cn (H.Z.); llyuxx@stu.sicau.edu.cn (Y.L.); 2College of Veterinary Medicine, Jiangsu Agri-Animal Husbandry Vocational College, Taizhou 225300, China; 2002010183@jsahvc.edu.cn; 3Department of Pharmacology, School of Basic Medical Sciences, Wuhan University, Wuhan 430071, China

**Keywords:** dairy cow placenta, antioxidant peptides, virtual screening, chromatography, oxidative stress

## Abstract

Our preliminary study identified dairy cow placenta extract (CPE) as a mixture of peptides with potent antioxidant activity both in vivo and in vitro. However, the specific antioxidant peptides (AOPs) responsible for this activity were not yet identified. In the current study, we employed virtual screening and chromatography techniques to isolate two peptides, ANNGKQWAEVF (CP1) and QPGLPGPAG (CP2), from CPE. These peptides were found to be less stable under extreme conditions such as high temperature, strong acid, strong alkali, and simulated digestive conditions. Nevertheless, under normal physiological conditions, both CP1 and CP2 exhibited significant antioxidant properties, including free-radical scavenging, metal chelating, and the inhibition of lipid peroxidation. They also up-regulated the activities of intracellular antioxidant enzymes in response to hydrogen-peroxide-induced oxidative stress, resulting in reduced MDA levels, a decreased expression of the Keap1 gene and protein, and increased levels of the Nrf2 and HO-1 genes and proteins. Furthermore, CP1 demonstrated superior antioxidant activity compared to CP2. These findings suggest that CP1 and CP2 hold potential for mitigating oxidative stress in vitro and highlight the efficacy of virtual screening as a method for isolating AOPs within CPE.

## 1. Introduction

Antioxidant peptides (AOPs) are biologically active peptides composed of 2–20 amino acids, known for their antioxidative properties. These peptides exhibit the ability to scavenge free radicals, trigger the body’s antioxidant pathways, enhance the expression of antioxidant enzymes, and mitigate oxidative stress within the body [1,2,3]. Zou et al. [4] have demonstrated that AOPs can be sourced from a variety of origins, primarily from the protein extracts of animals, plants, and microorganisms. The prevalent methods for isolating AOPs from protein extracts involve virtual screening and chromatography techniques. Virtual screening relies on computerized high-throughput analysis to identify AOPs with high activity and affinity based on the polypeptide sequences present in the protein extracts [5]. It is the most mature and widely studied method in terms of its mechanism of action and the material basis of the drug treatment of diseases [6]. This method is less restricted by conditions, allowing you to perform high-throughput screenings of large numbers of peptides while avoiding the disadvantages of traditional methods, i.e., being time-consuming and expensive, in screening bioactive ingredients [7]. However, the basic idea of this method is statistics, which requires more known information, so the matching results may lack accuracy [8]. At the same time, it requires comprehensive and reasonable sources of drug molecules and targets for research work, and many issues that may be ignored need to be considered, such as the actual content of the drugs and the influence of different processing methods on drug efficacy. On the other hand, chromatography separates AOPs based on their distinct physicochemical properties, driven by the varying partition coefficients of peptides in hydrolysates present in different column stationary phases and mobile phases. It has the characteristics of a good separation effect and convenient operation, as well as having various methods of adoption. However, it has high requirements for sample purity and poor qualitative effect, and the structural integrity of the peptides may be damaged during the separation process [9], resulting in inaccurate separation results. Furthermore, chromatography is known to be laborious, expensive, and time-consuming [7]. Chemical synthesis has become a practical technology to study the structure and function characterization of proteins. At present, solid-phase peptide synthesis technology is relatively mature and can be synthesized according to the length, purity, and sequence of peptides. It has the advantages of high efficiency and simple operation and is suitable for the synthesis of short and medium peptides [10].

In vitro and in vivo assays are the primary methods for studying the antioxidant activity of AOPs. In vitro assays include chemical and cell modeling methods. The chemical method accurately reflects the antioxidant activity of AOPs by measuring its free-radical scavenging rate, lipid peroxidation inhibition rate, transition metal ion chelating rate, and reducing power using chemical reagents. This method is easy and fast to operate [4,11,12,13], but Liu [14] showed that its biological relevance is low. The cell modeling method demonstrates the protective effect of AOPs on oxidative stress cells by establishing a cell model of oxidative stress and detecting the effects of AOPs on intracellular reactive oxygen species, malondialdehyde (MDA), antioxidant enzymes, and antioxidant pathways [13]. In vivo assays, on the other hand, are used to detect relevant antioxidant indexes and assess AOPs’ bioavailability by establishing an animal model of oxidative damage [13]. These assays are costly and have a complicated testing cycle [15]. Therefore, in vitro methods are primarily utilized to study the antioxidant activity of AOPs.

Placenta is a kind of traditional Chinese medicine, known as “Ziheche”, that has the function of tonifying qi and blood, nourishing the liver and kidney. Dairy cow placenta is a rich natural resource with a large mass [16]. Its composition is similar to that of human placenta, but it has not been effectively utilized. Dairy cow placenta extract (CPE) comprises polypeptides with a molecular weight below 3100 Da and exhibits potent antioxidant activity [17]. In addition, our previous studies have shown that CPE can enhance the activities of catalase (CAT), glutathione peroxidase (GSH-Px), peroxidase (POD), superoxide dismutase (SOD), and glutathione levels in the serum and liver of mice within an aging model. These effects contribute to a reduction in MDA and levels of reactive oxygen species, resulting in the inhibition of oxidative stress and the postponement of skin and liver aging [18,19]. Nonetheless, the specific AOP component of CPE responsible for its antioxidant properties remains ambiguous. In light of this, the present study employs virtual screening and chromatography to isolate AOPs within CPE, assess and compare their antioxidant stability and efficacy, examine their structure, and determine the optimal isolation method. This study laid a theoretical foundation for the development and utilization of cow placenta, clarified the antioxidant components of CPE, and improved the bioavailability of CPE, while providing a methodological framework for the isolation of AOPs from other animal sources.

## 2. Materials and Methods

### 2.1. Materials and Reagents

The preparation of CPE was carried out with reference to previous research methods [19]. Papain was used to hydrolyze the dairy cow placenta under the optimal conditions, separate the supernatant after centrifugation, and then freeze dry. MS technology was used to identify the proteins and peptides in the products; this was repeated three times, and the specific operation referred to the previously reported method. The relative molecular weight of the peptides in CPP ranged from 800 to 3100 Da, and the amino acid sequence comparison shows that the reducing ability of CPP is mainly provided by isoleucine–arginine, glutamine–leucine, leucine–lysine and other residues [17,20].

RAW264.7 cells were donated by the Pathology Laboratory of Sichuan Agricultural University (Chengdu, China); peptides ANNGKQWAEVF and QPGLPGPAG were synthesized by Sangon Biotech Co., Ltd. (Shanghai, China); 30DEAE filler was purchased from Suzhou Nanomicro Technology Co., Ltd. (Suzhou, China); Sephadex G25 packing was purchased from Rhawn (Shanghai, China); DPPH assay kit was purchased from Shanghai Yuanye Bio-Technology Co., Ltd. (Shanghai, China); MTS assay kit was supplied by Promega (Madison, WI, USA); SOD ELISA kits, CAT ELISA kits, GSH-Px ELISA kits, and MDA ELISA kits were purchased from Enzyme Link Bio (Shanghai, China); RAW264.7 special medium CM-0190 was purchased from Wuhan Pricella Biotechnology Co., Ltd. (Wuhan, China).

### 2.2. Virtual Screening Method to Isolate AOP

All peptide sequences in CPE were identified from a previous study [17]. Referring to previous studies [21,22,23,24], various websites (Table 1) were used to predict their antioxidant activity, stability, water solubility, sensitization, and toxicity. Through this comprehensive analysis, the top 5 peptides in terms of their physicochemical properties were screened as potential AOPs.

The molecular docking method was utilized to further screen AOPs from dairy cow placenta. Initially, the 3D structure of the potential AOPs was created using ChemDraw 19.0 software, imported into Discovery Studio 2019 client software, and processed with the removal of hydrogen atoms. Subsequently, the structure was utilized as a ligand, with TX6 (Pub Chem ID: 121488089) serving as the positive control. Following this, the crystal structure of the Keap1 protein (PDB ID: 2FLU) was obtained from the RCSB Protein Data Bank (https://www.rcsb.org (accessed on 24 July 2024)) and imported into the Discovery Studio 2019 client software. The Keap1 protein was prepared as a receptor after removing its ligands and water, and it was hydrogenated and cleaned. The docking procedure was then carried out by selecting the active center (x: 5, y: 9, z: 1, radius: 10 Å) and applying the “-CDOCKER” protocol. Lastly, the AOP was filtered based on the -CDOCKER_INTERACTION_ENERGY(-CIE) ranking in the docking results, denoted as CP1.

### 2.3. Chromatography to Isolate AOPs

#### 2.3.1. Ion Exchange Chromatography

A total of 25 mL of 30DEAE cellulose ion exchange resin packing was loaded into a 2 × 20 cm glass chromatography column, and the equilibrium column was washed with phosphate buffer (pH = 7.4) of 3 times the column volume. A CPE solution (5 mL, 50.0 mg/mL) was taken and added to a 30DEAE ion-exchange chromatography column (15 × 250 mm). The column was initially equilibrated with PBS and then eluted in a stepwise manner using PBS (pH = 8.0) and NaCl solutions at concentrations of 0.2, 0.4, and 0.6 mol/L, with a flow rate set at 2 mL/min. The eluate was collected in 2 mL fractions per tube and analyzed at a wavelength of 280 nm.

#### 2.3.2. Molecular Sieve Chromatographic Separation

Fractions were determined based on absorbance, and the DPPH radical scavenging activity of each fraction was measured. The most active fraction solution (1 mL, 20 mg/mL) was selected and added to a Sephadex G25 molecular sieve chromatography column (15 × 250 mm), pre-equilibrated using ultrapure water, and eluted stepwise with ultrapure water at a flow rate of 2 mL/min. The eluent (2 mL/tube) was collected and detected at 280 nm.

#### 2.3.3. RP-HPLC Separation

The fractions were determined based on absorbance, and the DPPH radical scavenging activity of each fraction was measured. The fraction solution with the strongest activity (50 μL, 5 mg/mL) was selected and added to a C18 column (4.6 × 250 mm) for further purification using a Waters 2695 high-performance liquid chromatograph. The linear gradient elution of 2% acetonitrile in mobile phase A and 98% acetonitrile (5–8%, 0–5 min; 8–18%, 5–40 min) in mobile phase B was carried out at a flow rate of 0.7 mL/min and detected at 214 nm.

#### 2.3.4. LC–MS/MS Identification

The DPPH radical scavenging capacity of each fraction was collected and measured, and the most active fraction solution was applied to LC–MS/MS for sequence identification. An easy-nLC high-performance liquid chromatograph in tandem with a Q-Exactive mass spectrometer was used to identify the polypeptide sequence in the strongest antioxidant fraction obtained in the preceding step. The fraction with the strongest antioxidant activity obtained from the above step was dissolved in 0.1% formic acid aqueous solution and used as the sample solution. Conditions for liquid chromatography: A C18 reverse phase column (0.15 mm × 150 mm, 3 μm) was used as the analysis column, with 0.1% formic acid aqueous solution as mobile phase A and 0.1% formic acid and 80% acetonitrile aqueous solution as mobile phase B. Linear gradient elution was adopted after sample loading at the flow rate of 600 nL/min. Elution conditions: 0–2 min, mobile phase B—4–8%; 2–45 min, mobile phase B—8–28%; 45–55 min, mobile phase B—28–40%; 55–56 min, mobile phase B—40–95%. The mass spectrum analysis conditions are as follows: primary mass spectrum parameters—resolution is 70,000, AGCtarget is 3 × 10^6^, MaximumIT is 100 ms, and Scanrange is 100 to 1500 *m*/*z*; secondary mass spectrum parameters—resolution 17,500, AGCtarget 1 × 10^5^, MaximumIT 50 ms, TopN 20, and NCE/steppedNCE 28. Finally, the polypeptide with the highest abundance in the identification results was selected, which was denoted as CP2. 

### 2.4. Synthesize CP1 and CP2

CP1 and CP2 were synthesized by Sangon Biotech (Shanghai) Co., Ltd. (Shanghai, China), with purity greater than 95%.

### 2.5. In Vitro Assay of CP1 and CP2 Antioxidant Activities

The hydroxyl radical scavenging rates of CP1 and CP2 were assayed with reference to the method of Fu et al. [25].

The DPPH radical scavenging rates of CP1 and CP2 were assayed with reference to the method of Hogan et al. [26].

The transition metal chelation rates of CP1 and CP2 were detected with reference to the method of Gu et al. [27].

The inhibition of the lipid peroxidation of CP1 and CP2 was assayed with reference to the method of Shen et al. [19].

### 2.6. Establishment of the H_2_O_2_-Induced Oxidative Stress RAW264.7 Cell Model

RAW264.7 cells were cultured in a 5% CO_2_ environment at 37 °C using a specific complete medium (DMEM (PM150210) + 10% FBS (164210-50) + 1%P/S (PB180120)) designed for RAW264.7 cells. Cells in the logarithmic growth phase were harvested and seeded into 96-well plates at a density of 1 × 10^5^ cells/mL per well with 100 μL of inoculation volume. Subsequently, the plates were pre-cultured for 12 h at 37 °C in a 5% CO_2_ atmosphere, following which the medium was removed.

The pre-cultured RAW264.7 cells were then divided into control (C) and model (M) groups. Then, 100 μL of complete medium was added to group C, and 100 μL of complete medium containing concentrations of 250, 300, 350, 400, and 450 μmol/L H_2_O_2_ was added to group M. After continuing the incubation for 6 h, the medium was discarded. Cell viability was assessed using the MTS method. The optimal H_2_O_2_ concentration for inducing the oxidative stress model was determined when cell viability decreased to 50%, ensuring that the cells maintained a specific viability level while meeting the damage criteria [3].

### 2.7. Effects of CP1 and CP2 on the Survival of Normal and Oxidatively Stressed Cells

Pre-cultured RAW264.7 cells were allocated into groups C, CP1, and CP2. Group C received 100 μL of complete medium, while groups CP1 and CP2 were supplemented with 100 μL of complete medium containing CP1 and CP2, respectively, at final concentrations of 1.25, 2.5, 5, 10, 20, and 40 μg/mL. The cells were then incubated for 24 h before discarding the medium. Cell viability was assessed using the MTS method to identify non-toxic concentration ranges for subsequent experiments.

In another set of experiments, pre-cultured RAW264.7 cells were divided into groups C, M, CP1, and CP2. Group C and M were supplemented with 100 μL of complete medium, while groups CP1 and CP2 received 100 μL of complete medium containing CP1 and CP2, respectively, at final concentrations of 1.25, 2.5, 5, 10, and 20 μg/mL. After a 24 h incubation period, the medium was removed. Group C received 100 μL of complete medium, while groups M, CP1, and CP2 were treated with 100 μL of complete medium containing 400 μmol/L H_2_O_2_ for 6 h. Subsequently, cell viability was determined using the MTS method. The optimal concentrations of CP1 and CP2 that maximized cell viability were chosen for further studies on their preventive effects against oxidative stress.

### 2.8. RAW264.7 Detection of Cellular Antioxidant Enzymes and MDA

RAW264.7 cells were seeded into 6-well plates at a density of 1 × 10^6^ cells/mL per well with an inoculation volume of 2 mL and incubated in pre-culture for 12 h at 37 °C with 5% CO_2_. Pre-cultured RAW264.7 cells were divided into groups C, M, CP1, and CP2. Groups C and M received 2000 μL of complete medium, while groups CP1 and CP2 were treated with 2000 μL of complete medium containing 20 μg/mL of CP1 and CP2, respectively. The incubation continued for 24 h. Subsequently, the medium was replaced. Group C received 2000 μL of complete medium, while groups M, CP1, and CP2 were exposed to 400 μmol/L H_2_O_2_ in 2000 μL of complete medium for 6 h. After the incubation period, the medium was removed, and the cells were washed three times with PBS. Intracellular levels of SOD, CAT, GSH-Px, and MDA were measured using ELISA kits following the manufacturer’s instructions.

### 2.9. RAW264.7 Detection of Relative Expression of Keap1, Nrf2, and HO-1 Genes in Cells

Real-time fluorescence quantitative PCR was utilized to assess the relative expression levels of *Keap1*, *Nrf2*, and *HO-1* genes in the cells obtained. Following RNA extraction from the cell samples, reverse transcription and quantitative PCR analyses were carried out. The reaction conditions consisted of pre-denaturation at 95 °C for 30 s, denaturation at 95 °C for 15 s, and annealing/extension at 60 °C for 30 s, with a total of 40 cycles. 

The Primer sequences for *Keap1*, *Nrf2*, *HO-1*, and *GAPDH* genes are shown in Appendix A. Analysis was performed by the 2^−ΔΔCt^ method.

### 2.10. RAW264.7 Detection of Relative Expression of Keap1, Nrf2, and HO-1 Proteins in Cells

Protein immunoblotting was performed to assess the relative expression levels of Keap1, Nrf2, and HO-1 proteins in the cells collected. The cell samples were lysed using RIPA lysis buffer, and 50 μg of the lysates was separated by 10% SDS-PAGE, and then transferred onto PVDF membranes. Each membrane was preincubated for 1.5 h at room temperature in Tris-buffered saline with 0.05% Tween 20 and 5% skimmed milk at a pH of 7.6. Subsequently, specific antibodies against Keap1, Nrf2, HO-1, and β-actin were individually applied to the PVDF membranes. Immunoreactive bands were visualized by incubation with the respective secondary antibodies and ECL chemiluminescent reagents. β-actin served as the internal reference. After capturing images, the gray value of the bands was quantified relative to the internally referenced gray value.

### 2.11. Detecting CP1 and CP2 Stability

We refer to Chai et al. [28] and Gallego et al. [29] methods to evaluate the antioxidant stability of AOPs at different temperatures, pH levels, and simulated digestion conditions.

### 2.12. Statistical Analysis

The test results were expressed as mean (M) ± standard deviation (SD). SPSS 26.0 software was used for ANOVA, and Dunnett’s T3 method was used for the significance test. *p* < 0.01 meant that the difference was extremely significant, *p* < 0.05 meant that the difference was significant, and 0.05 < *p* < 0.1 meant that there was a trend of difference. Graphs were generated using GraphPad Prism 9.0.0 software.

## 3. Results

### 3.1. Virtual Screening Method to Isolate CP1

The CPE comprised 129 peptides, which were assessed for antioxidant activity, water solubility, stability, sensitization, and toxicity. The analysis revealed that 68.75% of the peptides exhibited antioxidant activity, 43.75% demonstrated good stability, 68.75% displayed good water solubility, 53.91% were non-sensitizers, and all of the peptides were non-toxic (Appendix A). The top five peptides, selected based on their physicochemical properties, were identified as potential AOP candidates for molecular docking (Table 2). A comparative analysis with known functional bioactive peptides in the BIOPEP-UWM database (https://biochemia.uwm.edu.pl/biopep-uwm/, accessed on 13 September 2023.) confirmed that these five peptides were novel bioactive peptides.

Molecular docking was utilized to determine the binding affinity of AOPs to Keap1 and to screen AOPs. As shown in Table 3, the -CIE values of the five peptides were 109.87, 109.32, 105.85, 105.81, and 51.94 kJ/mol, respectively, which were higher than the -CIE of the positive control TX6 (25.58 kJ/mol), indicating that all five peptides had a higher affinity for Keap1 than TX6, and ANNGKQWAEVF demonstrated the highest affinity. From the docked 3D-structure picture, ANNGKQWAEVF can compete with Nrf2 for the binding site by occupying the region above the central cavity of the Keap1 Kelch structural domain [30] (Figure 1A); in the docked 2D structure, it is known that ANNGKQWAEVF can form a hydrogen bond interaction force with 13 amino acid residues such as TYR334, ALA366, and ARG380; a hydrophobic interaction force with amino acid residues LEU557, IE559, and TYR572; an electrostatic interaction force with amino acid residue ARG415; and occupies three key sites (Figure 1B). This indicates the stable binding of ANNGKQWAEVF to Keap1. Therefore, ANNGKQWAEVF (H-Ala-Asn-Asn-Gly-Lys-Gln-Trp-Ala-Glu-Val-Phe-OH) was designated as CP1, which has a molecular formula of C_57_H_82_N_16_O_17_ and a molecular weight of 1263.38 Da.

### 3.2. Chromatography to Isolate CP2

As shown in Figure 2, four fractions were obtained by the ion exchange chromatography separation of CPE (Figure 2A), of which the D2 fraction had the strongest DPPH radical scavenging activity of 15.2 ± 0.24% (5 mg/mL) (Figure 2B). Two fractions were obtained by the molecular sieve chromatography separation of the D2 fraction (Figure 2C), of which the G1 fraction had the strongest DPPH radical scavenging activity of 23.58 ± 0.33% (5 mg/mL) (Figure 2D). Three fractions were obtained by the RP-HPLC separation of the G1 fraction (Figure 2E), of which the R2 fraction exhibited the highest DPPH radical scavenging activity of 10.35 ± 0.15% (0.1 mg/mL) (Figure 2F). The R2 fraction was identified by LC–MS/MS, and 21 peptides were found (Appendix A). Among these peptides, QPGLPGPAG emerged as the most abundant. Consequently, QPGLPGPAG (H-Gln-Pro-Gly-Leu-Pro-Gly-Pro-Ala-Gly-OH) was chosen to be noted as CP2, which has a molecular formula of C_35_H_56_N_10_O_11_, a molecular weight of 792.89 Da, and an extraction rate of approximately 10%. The secondary mass spectra of CP2 are depicted in Figure 2G.

### 3.3. Purity Analysis of Synthetic AOPs

After the solid-phase synthesis of CP1 and CP2, its liquid chromatogram and mass spectrometry are shown in Figure 3. The highest purity components of the synthetic products are CP1 and CP2, and the purity is greater than 95%.

### 3.4. In Vitro Antioxidant Activity of CP1 and CP2

The IC_50_ values corresponding to the antioxidant activity assays of CP1 and CP2 were calculated and are presented in Table 4. The IC_50_ values for the hydroxyl radical scavenging rates were 1.71 mg/mL for CP1 and 0.85 mg/mL for CP2. Similarly, the IC_50_ values for the DPPH radical scavenging rates were 1.34 mg/mL for CP1 and 1.00 mg/mL for CP2. Additionally, the IC_50_ values for the transition metal chelating rates were 5.20 mg/mL for CP1 and 4.67 mg/mL for CP2, while the IC_50_ values for the lipid peroxidation inhibition rates were 3.90 mg/mL for CP1 and 3.49 mg/mL for CP2.

### 3.5. Effect of H_2_O_2_, CP1, and CP2 on the Survival of RAW264.7 Cells

The survival rate of RAW264.7 cells is illustrated in Figure 4. Upon inducing the oxidative stress model in RAW264.7 cells with H_2_O_2_, the cell survival rate decreased as the concentration of H_2_O_2_ increased. Following exposure to a concentration of 200 μmol/L H_2_O_2_, the cell activity exceeded 90%, showing no significant difference from the control group (*p* > 0.05). At a concentration of 250 μmol/L, the cell survival rate was 83.51 ± 6.15%, which was significantly lower than the control group (*p* < 0.05). Subsequently, at concentrations of 300, 350, 400, and 450 μmol/L, the cell survival rates were 79.06 ± 3.21%, 64.73 ± 0.99%, 48.68 ± 5.48%, and 26.10 ± 1.65%, respectively, all highly significantly lower than the control group (*p* < 0.01) (Figure 4A). As the cell survival rate approached 50% after exposure to 400 μmol/L H_2_O_2_, this concentration was deemed the optimal choice for inducing the oxidative stress model.

Following the incubation of RAW264.7 cells with CP1 and CP2, the cell viability did not show a significant difference compared to group C at concentrations ranging from 1.25 to 20.00 μg/mL (*p* > 0.05). However, at a concentration of 40.00 μg/mL, the cell viability was significantly lower than that of group C (*p* < 0.01). The absence of any toxic effects on RAW264.7 cells at concentrations between 1.25 and 20 μg/mL (Figure 4B) led to the selection of this concentration range for subsequent experiments.

After pre-treating oxidatively stressed RAW264.7 cells with CP1 and CP2, the cell survival rate increased proportionally with the concentration of CP1 and CP2. Specifically, at concentrations of 2.5 and 5 μg/mL, the cell survival rates for the CP1 group were 58.98 ± 3.05% and 62.31 ± 2.38%, respectively, which were significantly higher than those of group M (*p* < 0.01). At a concentration of 10 μg/mL, the cell survival rates were 64.54 ± 2.05% and 65.28 ± 1.81%, respectively, which was, again, significantly higher than group C (*p* < 0.01). The cell survival rate peaked at 20.00 μg/mL, with rates of 76.37 ± 4.49% and 70.14 ± 2.26%, which was significantly higher than those of group C (*p* < 0.01) (Figure 4C). Considering the maximum cell viability observed at 20.00 μg/mL, this concentration was deemed optimal for the protective effects of CP1 and CP2.

### 3.6. Effects of CP1 and CP2 on Antioxidant Enzymes and MDA in Oxidatively Stressed RAW264.7 Cells

As shown in Figure 5, after H_2_O_2_ induction, RAW264.7 intracellular antioxidant enzymes SOD, CAT, and GSH-px activities were all significantly decreased, and MDA levels were significantly increased (*p* < 0.01). After pretreatment with CP1 and CP2, the intracellular SOD levels were 75.107 ± 6.87 ng/mL and 77.43 ± 5.03 ng/mL, respectively, which were significantly higher than the group M level of 61.37 ± 6.40 ng/mL (*p* < 0.01) (Figure 5A); CAT levels were 1386.22 ± 92.26 pg/mL and 1481.94 ± 46.27 pg/mL, which were significantly higher compared with group M (*p* < 0.01) (Figure 5B); GSH-Px levels were 78.50 ± 5.61 ng/mL and 82.49 ± 11.09 ng/mL, which were both significantly higher compared with group M (*p* < 0.05) (Figure 5C); and MDA levels decreased to 27.50 ± 2.42 nmol/mL (*p* < 0.05) and 25.54 ± 2.74 nmol/mL (*p* < 0.01) (Figure 5D). In addition, CAT activity was significantly higher in the CP1 group than in the CP2 group (*p* < 0.05) (Figure 5C).

### 3.7. Effects of CP1 and CP2 on Antioxidant Pathways in Oxidatively Stressed RAW264.7 Cells

In Figure 6, post-H_2_O_2_ treatment revealed a significant increase in Keap1 gene and protein expression (*p* < 0.01), while Nrf2 and HO-1 gene and protein expression were markedly decreased (*p* < 0.01). Conversely, pretreatment with CP1 and CP2 led to a high significant reduction in Keap1 gene expression (*p* < 0.01) (Figure 6A) and a substantial increase in Nrf2 and HO-1 gene expression (*p* < 0.01) (Figure 6B,C). Moreover, CP1 and CP2 pretreatment significantly decreased Keap1 protein expression (*p* < 0.01) and notably increased Nrf2 and HO-1 gene expression (*p* < 0.01) (Figure 6D,E), which was consistent with the gene expression results. Additionally, the relative expression of Nrf2 gene in the CP1 group was significantly higher than in the CP2 group (*p* < 0.05), while the relative expression of HO-1 gene was substantially higher in the CP1 group compared with CP2 (*p* < 0.01). The relative expression of Keap1 protein in the CP1 group was significantly higher than in the CP2 group (*p* < 0.05), and the relative expression of HO-1 protein was substantially higher in the CP1 group than in the CP2 group (*p* < 0.01).

### 3.8. Stability of CP1 and CP2

The stability results of CP1 and CP2 are depicted in Figure 7. The free-radical scavenging rates of CP1 and CP2 exhibited significant variations at different temperatures (*p* < 0.05). Specifically, at temperatures ranging from 25 to 50 °C, 50 to 70 °C, and 70 to 90 °C, the free-radical scavenging rates of CP1 decreased by 18.28%, 52.32%, and 39.60%, while those of CP2 decreased by 8.60%, 26.97%, and 49.92%, respectively (Figure 7A). Moreover, the free-radical scavenging rates of CP1 and CP2 differed significantly at various pH levels (*p* < 0.05), with the exception of CP2 at pH 3, showing no significant difference (*p* > 0.05) compared to pH 9. Specifically, the free-radical scavenging rates of CP1 decreased by 34.00%, 40.60%, 72.41%, and 34.17% at pH 7–5, 5–3, 7–9, and 9–11, respectively, whereas CP2 exhibited decreases of 12.52%, 39.96%, 43.89%, and 31.25% under the same pH conditions (Figure 7B). Additionally, in gastrointestinal (G) digestion, CP1 and CP2 demonstrated significantly lower free-radical scavenging rates compared to the control group (*p* < 0.01), decreasing by 82.31% and 79.21%, respectively. Notably, there was no significant difference in the free-radical scavenging rates between gastrointestinal (GI) digestion and G digestion for CP1 and CP2 (*p* > 0.05), with reductions of 38.64% and 34.01%, respectively (Figure 7C). These findings suggest that the antioxidant stability of CP1 and CP2 notably decreased under varied temperatures, pH levels, and digestion conditions, which was potentially attributed to the disruption of their spatial structures or amino acid compositions. Notably, the decrease in stability of CP2 was approximately half of that observed for CP1 at temperatures of 50–70 °C and pH levels 5 and 9, indicating a comparatively stronger stability of CP2 compared to CP1 under these specific conditions.

## 4. Discussion

### 4.1. Effects of Separation Method on the Separation of AOPs

Virtual screening and chromatography are frequently employed techniques for the isolation of AOP. Virtual screening involves computerized analysis utilizing network techniques to identify peptides with antioxidant properties as ligands, followed by molecular docking to screen for AOP [5]. Conversely, chromatography is a conventional isolation method that purifies AOP through the stepwise fractionation of enzyme digests to obtain fractions exhibiting the highest antioxidant activity [11].

The advantage of the virtual screening method is that it is easy to operate and can screen AOPs in high-throughput [5,30]. However, it may yield inaccurate results due to the limited accuracy of the 3D conformation of ligands and receptors during molecular docking, as well as the use of approximation scoring functions [8]. Notably, the accuracy of molecular docking results can be increased by selecting the peptide with antioxidant potential and a stable nature as the molecularly docked ligand after adopting the network analysis technique [31]. Zou et al. [4] employed this method to screen 129 peptides with favorable properties, predicting bioactivity, water solubility, human intestinal absorption, and toxicity among 2499 peptides from pork intestinal hydrolysate. They successfully isolated 15 AOPs. In a similar vein, Zhang et al. [32] identified two AOPs from 109 peptides derived from dried coconut flour hydrolysate through virtual and molecular docking screenings, all showing significant antioxidant activity. Virtual screening was utilized to predict the antioxidant potential, stability, water solubility, toxicity, and sensitization of 129 peptides in CPE, and five peptides were obtained by screening and molecularly docked with Keap1 receptor proteins to increase the accuracy of the isolation results based on the docking results. CP1 was determined to be ANNGKQWAEVF.

Wang et al. [33] demonstrated the efficacy of chromatography in utilizing the free-radical scavenging rate as a screening index for fractions with antioxidant activity. This approach ensures that the isolated AOPs exhibit antioxidant properties and underscores the reliability of the results. In contrast, Singh and Bharadvaja [7] highlighted the laborious nature of the process, noting the potentially high number of polypeptides in the fractionated products, which complicates the screening of target AOPs. Li et al. [34] illustrated the application of chromatography in separating AOPs from milk fat globule membrane hydrolysate using DEAE, leading to the identification of 497 peptides from the fractionated products. Ren et al. [35] detailed the separation of AOPs from broken rice hydrolysate through Sephadex G-25 and FPLC methods, resulting in the identification of 98 peptides from the fractionated products, thereby increasing the screening complexity of AOPs. To streamline the purification process and enhance purity, we employed reversed-phase high-performance liquid chromatography, ion exchange chromatography, and molecular sieve chromatography. This methodology facilitated the isolation of only 21 polypeptides in the final purified products, leading to the determination of CP2 as QPGLPGPAG based on the abundance of polypeptides.

In the comparison, between the methods of obtaining AOPs, it is suggested that there might be variations in the antioxidant capacity of AOPs acquired through these two distinct approaches. The virtual screening method, employing peptide affinity for Keap1 as a screening parameter, exclusively confirms the capability of CP1 to activate the Keap1/Nrf2 pathway. On the other hand, the chromatographic method, utilizing the free-radical scavenging rate as a screening metric, solely validates the ability of CP2 to scavenge free radicals. While the majority of AOPs exhibit both antioxidative pathway activation and free-radical scavenging abilities, there are instances where certain AOPs only possess one of these functionalities. Han et al. [36] demonstrated that tuna AOPs (LCGEC) displayed higher affinity for binding to Keap1 than TX6 through molecular docking technology. This led to the activation of the Keap1/Nrf2 pathway and increased antioxidant enzyme activity in oxidative stress HaCaT cells and healthy mice. However, its DPPH radical scavenging rate, IC_50_, exceeded 10 mg/mL, indicating a limited radical scavenging capacity. Wang et al. [37] isolated eleven AOPs from skipjack tuna protein extract using chromatography, among which SGE, QEAE, and QAEP exhibited DPPH free-radical scavenging IC_50_ values of 1.34, 1.77, and 1.11 mg/mL, respectively, signifying effective free-radical scavenging capabilities but an inability to alleviate Chang cell oxidative stress. Therefore, the synthesis of CP1 and CP2 in subsequent experiments is deemed necessary to verify their potential to scavenge free radicals and mitigate cellular oxidative stress.

### 4.2. Evaluation of the Antioxidant Activity of CP1 and CP2

AOP antioxidant activity is primarily assessed through chemical and cell modeling methods. The chemical method for evaluating AOP antioxidant activity can be categorized into two main types: hydrogen atom transfer and electron transfer. In assays such as hydroxyl and DPPH radical scavenging, AOP transforms free radicals into anions via electron transfer. Moreover, in the lipid peroxidation inhibition assay, AOP hinders lipid peroxidation by releasing hydrogen atoms, thereby deactivating free radicals [38]. Furthermore, Zheng et al. [39] demonstrated that AOP can counteract oxidative stress through its chelating capability by binding to transition metal ions and preventing their reaction with H_2_O_2_. Yang et al. [40] observed that duck blood AOP (EVGK) exhibited a DPPH radical scavenging rate of 26.63% at a concentration of 0.5 mg/mL and a transition metal chelation rate of 16.35%. Additionally, Han et al. [36] found that tuna AOP (LCGEC) displayed a DPPH radical scavenging rate of 16.83% at a concentration of 1 mg/mL, while Guo et al. [41] reported that sheepskin AOP (YGPEP) had an IC_50_ lipid peroxidation inhibition rate of 3.94 mg/mL. Moreover, He et al. [42] highlighted that the IC_50_ values for hydroxyl radical scavenging and DPPH radical scavenging were 2.47 mg/mL and 1.59 mg/mL, respectively, for yellow croaker AOP (YFLWP), signifying potent antioxidant properties suitable for pharmaceutical or food applications. The DPPH radical scavenging and transition metal chelation rates of CP1 and CP2 closely resembled those of EVGK at 0.5 mg/mL, with a higher DPPH radical scavenging rate than LCGEC at 1 mg/mL. However, the lipid peroxidation inhibition, hydroxyl radical scavenging, and DPPH radical scavenging IC_50_ values were lower for CP1 and CP2 compared to YGPEP and YFLWP, indicating that CP1 and CP2 demonstrated antioxidant activities similar to EVGK and superior to LCGEC, YGPEP, and YFLWP, thereby displaying robust antioxidant capabilities.

### 4.3. Comparative Analysis of the Antioxidant Activity of CP1 and CP2

The antioxidant activity of AOP is closely related to its molecular weight, amino acid composition, and sequence [33]. AOPs with molecular weights ranging from 500 to 1800 Da and comprising 2–20 amino acids are more adept at countering free radicals and exhibiting antioxidant effects [43]. An optimal presence of hydrophobic amino acids (Ala, Ile, Leu, Pro, Phe, Val, and Trp) facilitates AOP cell entry and enhances antioxidant activity. Conversely, an excess of hydrophobic residues diminishes AOP solubility, thereby reducing its antioxidant efficacy [44]. Maize-derived AOP (QQPQPW, 782.34 Da, with 50.00% hydrophobic amino acid residues) demonstrated robust free-radical scavenging, transition metal chelation, and lipid peroxidation inhibition [45]. Bacillus amyloliquefaciens AOPs (APKGVQGPNG, 924.01 Da, with 40% hydrophobic amino acid residues) activated the Keap1/Nrf2 pathway and shielded RAW264.7 cells from oxidative stress [46]. The antioxidant activity of CP1 and CP2 closely mirrored the findings of the aforementioned AOP studies. CP1 and CP2, with respective molecular weights of 1263.38 and 792.89 Da, containing 11 and 9 amino acids and 45.00% and 55.56% hydrophobic amino acid residues, exhibited strong antioxidant capabilities.

The enhanced antioxidant activity of CP1 compared to CP2 is attributed to its unique amino acid composition. Aromatic amino acids (Phe, Trp, and Tyr) within AOPs stabilize free radicals through electron transfer mechanisms [47], while acidic amino acids (Glu and Asp) deactivate free radicals by releasing protons [48]. Additionally, acidic and basic amino acid residues (Lys, Arg, and His) contribute to the augmentation of transition metal chelation abilities [49] Moreover, the presence of hydrophobic amino acids at the N- and C-termini has been linked to heightened antioxidant activity [50,51]. Chen et al. [52] showed the SWDNFFR from wine lees AOP contains more of the above specific amino acids and has stronger antioxidant activity than other wine lees AOP (WDWVGGR, FMFDGFR). CP1 (ANNGKQWAEVF) contains Phe, Trp, Glu, and Lys residues and has hydrophobic amino acid residues both at the N-terminal and C-terminal ends, whereas CP2 (QPGLPGPAG) was devoid of these specific amino acids. Therefore, CP1 was stronger than CP2 in free-radical scavenging, transition metal chelation, lipid peroxidation inhibition, and the inhibition of cellular oxidative stress.

In the context of AOP, the choice of separation method can significantly impact its antioxidant activity. When compared to conventional chromatography, utilizing the network analysis technique in virtual screening enhances the bioactivity, antioxidant potential, stability, water solubility, and overall efficacy of the resultant AOP. This method is more likely to yield AOPs with superior antioxidant properties. Additionally, the molecular docking approach uses the interaction between peptides and the Keap1 protein as a screening criterion for AOP selection. This process ensures a higher probability of binding to Keap1, subsequently activating the Keap1/Nrf2 pathway and providing cellular protection against oxidative stress [33]. During the isolation of AOP from pork intestines, peptides exhibiting favorable bioactivity, human intestinal absorption, and water solubility were prioritized for molecular docking studies. As a result, the antioxidant capacity of the obtained AOP (DWWGSTVR) surpassed that of the chromatographically purified AOP (SDEEVEH) [4,53]. Specifically, CP1, obtained through virtual screening, showcased heightened antioxidant efficacy, improved stability, water solubility, and strong binding affinity with the Keap1 protein. These attributes led to a notable reduction in Keap1 protein expression levels in oxidative-stress-induced RAW264.7 cells and a significant upregulation in Nrf2 gene expression, as well as an increased expression of HO-1 gene and protein, when compared to CP2.

After testing the antioxidant activity of CP1 and CP2 using an in vitro method and cellular modeling, it was found that CP1 has stronger antioxidant activity; therefore, a virtual screening method is a more preferable method to isolate AOPs from CPE.

## 5. Conclusions

Bioactive peptides are specific amino acid sequences with beneficial physiological effects, some of which are activated by extraction from parental proteins, and their production and application have great development value and application prospects. Virtual screening and chromatographic methods were employed to isolate AOPs from CPE, resulting in the identification of ANNGKQWAEVF and QPGLPGPAG. These AOPs exhibit noteworthy antioxidant properties, including free-radical scavenging, the inhibition of lipid peroxidation, metal chelating abilities, and reducing power. Additionally, they are capable of activating the Keap1/Nrf2 pathway in oxidatively stressed cells, thereby mitigating cellular oxidative stress. Among the identified AOPs, ANNGKQWAEVF demonstrated superior antioxidant activity, underscoring the efficacy of the virtual screening approach for AOP isolation from CPE. Notably, both ANNGKQWAEVF and QPGLPGPAG exhibited optimal antioxidant activity at 25 °C and pH = 7, making them suitable for processing and storage under these conditions. However, their antioxidant activity is diminished in digestive environments, rendering them unsuitable for oral consumption as food or pharmaceutical ingredients. Therefore, the antioxidant capacity of CP1 and CP2 still needs to be studied and tested more so that it can play a role in many fields, such as beauty and agriculture, and provide a reference for the secondary development and utilization of biological resources. In addition, the current extraction rate of AOPs isolated from CPE is low, which may lead to the inability of AOP to be widely studied and applied. In the future, we will further explore more efficient separation methods for AOP, so as to lay a good foundation for the development and utilization of CPE.

## Figures and Tables

**Figure 1 antioxidants-13-00913-f001:**
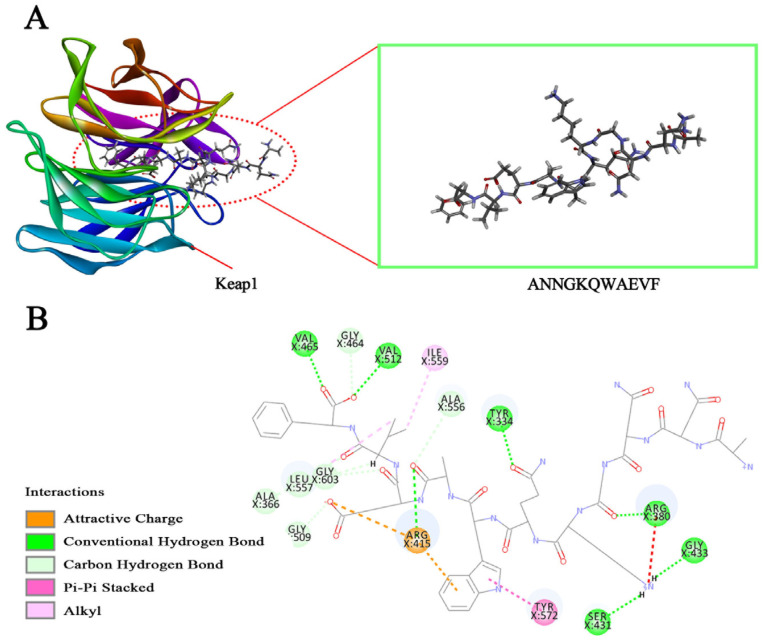
Schematic diagram of docking between ANNGKQWAEVF and Keap1 (PDB ID:2FLU). (**A**) is a 3D diagram of the interaction between polypeptide ANNGKQWAEVF and Keap1, and (**B**) is a 2D diagram.

**Figure 2 antioxidants-13-00913-f002:**
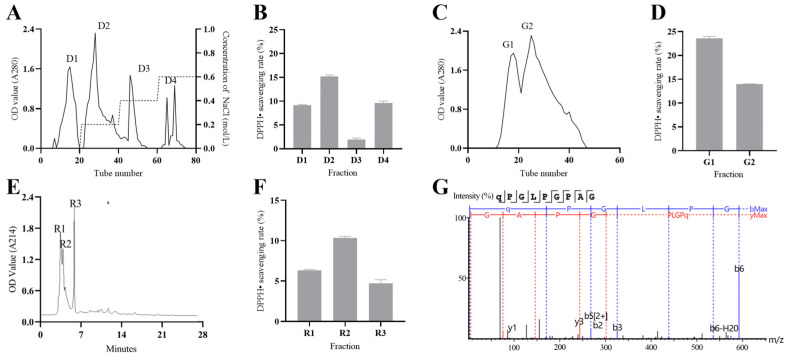
Chromatographic separation process and identification results. (**A**,**C**,**E**) show the elution curves of CPE, D2, and G1, respectively, while (**B**,**D**,**F**) show the DPPH free-radical clearance rate of corresponding fractions. (**G**) shows the secondary mass spectrum of polypeptide QPGLPGPAG, with b and y ions labeled by dashes.

**Figure 3 antioxidants-13-00913-f003:**
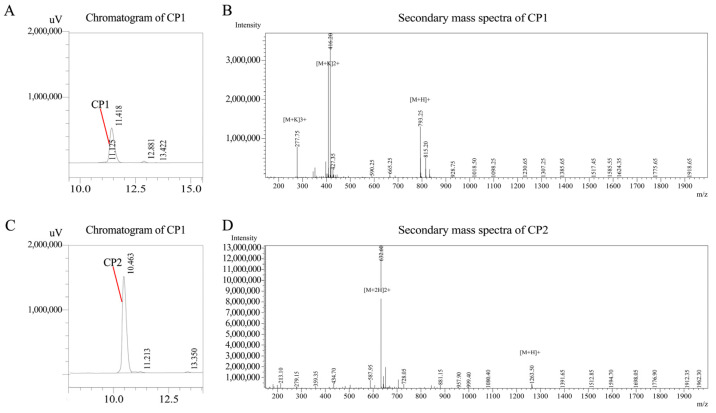
Liquid chromatography and secondary mass spectrometry of CP1 and CP2. (**A**,**B**) are liquid chromatograms of CP1 and CP2, respectively, with purity > 95%; (**C**,**D**) are the secondary mass spectra.

**Figure 4 antioxidants-13-00913-f004:**
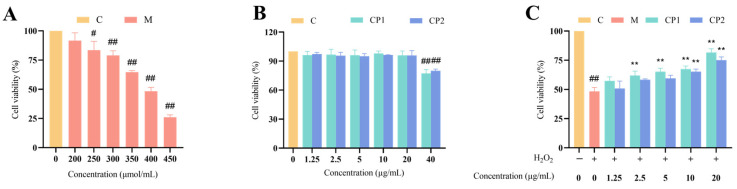
Survival rate of RAW264.7 cells. (**A**) shows the effect of H_2_O_2_ at different concentrations on the activity of RAW264.7 cells. (**B**) shows the effect of CP1 and CP2 on the survival rate of normal RAW264.7 cells. (**C**) shows the effect of CP1 and CP2 on the survival rate of RAW264.7 cells under oxidative stress. “#” and “##” represent *p* < 0.05 and *p* < 0.01, respectively, when compared with group C. When compared with group M, “**” *p* < 0.01. The same applies hereinafter.

**Figure 5 antioxidants-13-00913-f005:**
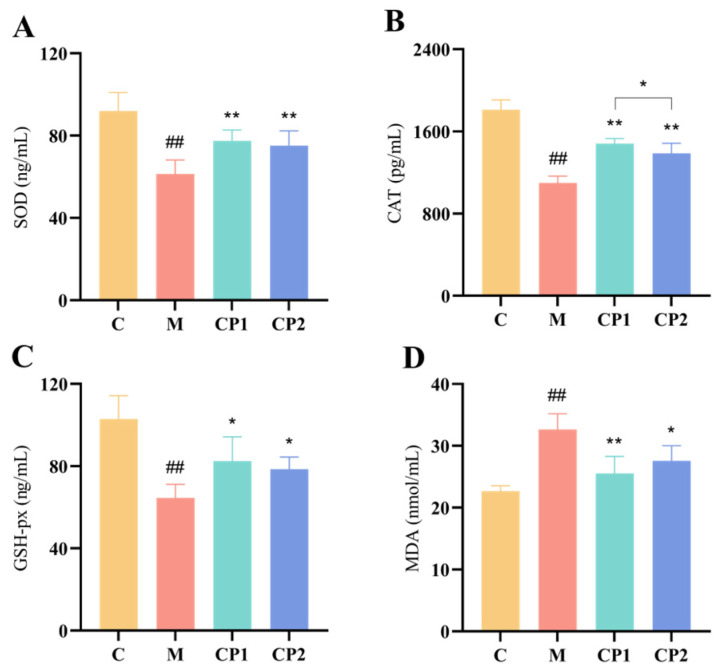
Effect of AOPs on antioxidant indicators of oxidative stress cells. (**A**–**D**) represent the SOD, CAT, GSH-px, and MDA levels of cells, respectively. “##” represent *p* < 0.01, when compared with group C. When compared with group M, “*” and “**” *p* < 0.05 and *p* < 0.01.

**Figure 6 antioxidants-13-00913-f006:**
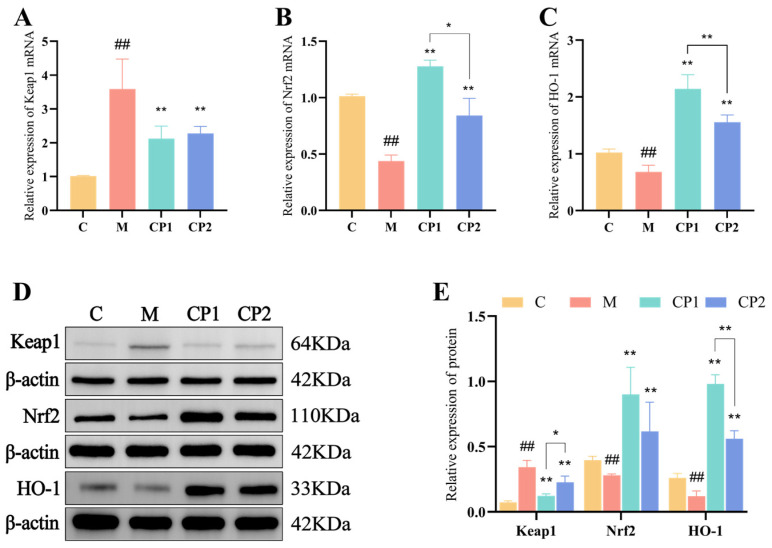
Effect of AOPs on antioxidant pathways of oxidative stress cells. (**A**–**C**) represent the relative mRNA expression levels of *Keap1, Nrf2*, and *HO-1*, respectively. (**D**,**E**) represent the relative expression levels of Keap1, Nrf2, and HO-1 proteins. “##” represent *p* < 0.01, when compared with group C. When compared with group M, “*” and “**” *p* < 0.05 and *p* < 0.01.

**Figure 7 antioxidants-13-00913-f007:**
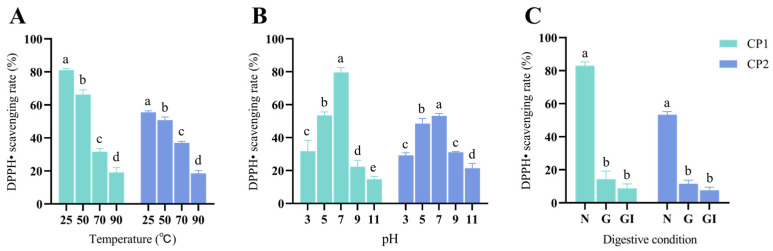
Stability of the CP1 and CP2. (**A**) represent free-radical scavenging rates of CP1 and CP2 at different temperatures. (**B**) represent free-radical scavenging rates of CP1 and CP2 at different pH. In (**C**), N represents the non-digestive phase, G represents the gastric digestive phase, and GI represents the gastrointestinal digestive phase. Those with different letter markers were marked as significant differences (*p* < 0.05), while those with the same letter markers were marked as insignificant differences (*p* > 0.05).

**Table 1 antioxidants-13-00913-t001:** Virtual screening website.

Physicochemical	Websites
Antioxidant activity	AnOxPP (http://www.cqudfbp.net/AnOxPP/index.jsp), accessed on 8 August 2023
Stability	ProtParam (http://web.expasy.org/protparam/), accessed on 13 August 2023
Water solubility	Innovagen (http://www.innovagen.com/proteomics-tools/), accessed on 26 July 2023
Sensitization	AllerTOP (v.2.0) (https://www.ddg-pharmfac.net/AllerTOP/), accessed on 30 July 2023
Toxicity	ToxinPred (http://crdd.osdd.net/raghava/toxinpred/), accessed on 8 August 2023

**Table 2 antioxidants-13-00913-t002:** Prediction table of biological and physical properties of peptides.

Peptide	Antioxidant Activity	Stability	Water Solubility	Sensitization	Toxicity
DLFENTNHTQVQ	+	−29.28	−1.23	–	–
ANNGKQWAEVF	+	−4.43	−0.78	–	–
WELTDDKNQRFF	+	−4.19	−1.51	–	–
RDNLLDDLQRLK	+	29.26	−1.27	–	–
DGRHDPRDDDLNLR	+	31.41	−2.29	–	–

**Table 3 antioxidants-13-00913-t003:** AOP molecular docking interaction ability.

Peptide	Molecular Weight	−CIE
ANNGKQWAEVF	1263.38 Da	109.87 KJ/mol
RDNLLDDLQRLK	1498.68 Da	109.32 KJ/mol
DGRHDPRDDDLNLR	1693.72 Da	105.85 KJ/mol
DLFENTNHTQVQ	1445.48 Da	105.81 KJ/mol
WELTDDKNQRFF	1598.70 Da	51.94 KJ/mol
TX6	331.40 Da	25.58 KJ/mol

**Table 4 antioxidants-13-00913-t004:** In vitro antioxidant activity of the CP1 and CP2.

Antioxidant Ability	CP1 (mg/mL)	CP2 (mg/mL)
Hydroxyl radical scavenging rate (IC_50_)	1.00	1.34
DPPH free-radical scavenging rate (IC_50_)	0.85	1.71
Transition metal chelation rate (IC_50_)	4.67	5.20
Lipid peroxidation inhibition rate (IC_50_)	3.49	3.90

## Data Availability

The data presented in this study are available in the article/Appendix A; further inquiries can be directed to the corresponding author.

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
