# Peer review of "Isolation and Characterization of Antioxidant Peptides from Dairy Cow (Bos taurus) Placenta and Their Antioxidant Activities"

_antioxidants, 2024, doi:10.3390/antiox13080913_

Round 1

Reviewer 1 Report

  • Materials and Methods:

    • Virtual Screening Details: Clarify the selection criteria for the top 5 peptides. Specify how the physicochemical properties were weighted in the selection process.
    • Chromatography Method: Provide a clearer step-by-step procedure for chromatography, including specific buffer compositions and pH values.
    • Cell Culture Conditions: Detail the composition of the "specific complete medium" used for culturing RAW264.7 cells.
  • Results:

    • Statistical Analysis: Ensure that all statistical tests are properly described, including justification for the choice of tests. Clarify the meaning of "high significance" (e.g., P < 0.01).
    • Graph Clarity: Figures should be clearer and more self-explanatory. Include legends and labels that are easy to read and understand without referring to the text.
  • Discussion:

    • Comparison of Methods: Provide a more balanced discussion on the advantages and limitations of virtual screening vs. chromatography. Include more references to previous studies that have used these methods.
    • Biological Relevance: Emphasize the biological relevance of the findings. Discuss the potential physiological impacts of CP1 and CP2 in vivo.
    • Conclusion: The conclusion should not only summarize findings but also suggest potential future research directions or applications of the isolated peptides.

This says nothing: "Acknowledgments: Authors express thanks to all persons who contributed to this study."

Mention the names.

Reviewer 2 Report

1.      In the article title, there should be a scientific name after cow indicating the species of the cow.

2. The authors ought to emphasize the importance of utilizing the CP (by-products). It is very critical to pinpoint and justify the research motivation. 

3. Line 80, the authors should give more information on the CPE. According to the previous study, lots of sample preparations were done before. However, it is important to provide the essential information regarding the sample integrity in terms of how it was prepared and handled, and the originality. 

4. Line 110, the authors mentioned “Chromatography” in section 2.3. It should be given more specific information on what kind of Chromatography was used. In addition, the authors addressed the use of LC-MS/MS in line 130 for sequence identification but the method of LC-MS/MS was not found in this manuscript. Strangely, the authors presented the results of the identified sequence in Figure 2G without providing the methodology. In Figure 2, it should be named Fig 2A, 2B….2G.

5.      Section 2.4, it is very sloppy to write limited information on the synthesized peptides CP1 and CP2. The authors should give more information on it.

6.      What were the yields of the CPE, CP1 and CP2? It is very important to know the yields because it is related to the production cost.

7.      If the authors can use computer tools to predict the AOP, why bother using the CPE to isolate the natural peptides? After prediction, it can be synthesized. The authors better to justify this in the manuscript.

Please refer to the comments above.

Round 2

Reviewer 2 Report

1. Editorial corrections need to be made. Check the comment below.

1. The newly added information at line 140, "25mL" and "2×20cm". There should be a space before the units. The authors are strongly recommended  to check the units for consistency throughout the whole manuscript. Same issues detected at section 2.3.4.

2. In section 2.3.4, there should be a reference for the LC-MS/MS method used.

3. In Antioxidant journal, the "Figures" are presented using the full name not the abbreviated term "Fig". Please correct them accordingly.

4. Still for the yields of the CPE, CP1 and CP2, the authors should address the difficulties and chanllenges in the manuscript. It is not acceptable to tell me the yield is small. This is not very scientific. The beauty of science is to share the true details discovered, discuss the issue and hopefully to make an improment out of it.    
